# Upconversion Luminescence Response of a Single YVO_4_:Yb, Er Particle

**DOI:** 10.3390/mi14051075

**Published:** 2023-05-19

**Authors:** Dmitry K. Zharkov, Andrey V. Leontyev, Artemi G. Shmelev, Larisa A. Nurtdinova, Anton P. Chuklanov, Niaz I. Nurgazizov, Victor G. Nikiforov

**Affiliations:** Zavoisky Physical-Technical Institute, FRC Kazan Scientific Center of RAS, Sibirsky Tract, 10/7, 420029 Kazan, Russia; dzharkov@list.ru (D.K.Z.); mailscrew@gmail.com (A.V.L.); sgartjom@gmail.com (A.G.S.); nurlari@yandex.ru (L.A.N.); a.chuklanov@gmail.com (A.P.C.); niazn@mail.ru (N.I.N.)

**Keywords:** upconversion particles, yttrium vanadate nanoparticles, upconversion luminescence, single particle spectroscopy, atomic-force microscopy, confocal optical microscopy

## Abstract

We present the results of the luminescence response studies of a single YVO_4_:Yb, Er particle of 1-µm size. Yttrium vanadate nanoparticles are well-known for their low sensitivity to surface quenchers in water solutions which makes them of special interest for biological applications. First, YVO_4_:Yb, Er nanoparticles (in the size range from 0.05 µm up to 2 µm), using the hydrothermal method, were synthesized. Nanoparticles deposited and dried on a glass surface exhibited bright green upconversion luminescence. By means of an atomic-force microscope, a 60 × 60 µm^2^ square of a glass surface was cleaned from any noticeable contaminants (more than 10 nm in size) and a single particle of 1-µm size was selected and placed in the middle. Confocal microscopy revealed a significant difference between the collective luminescent response of an ensemble of synthesized nanoparticles (in the form of a dry powder) and that of a single particle. In particular, a pronounced polarization of the upconversion luminescence from a single particle was observed. Luminescence dependences on the laser power are quite different for the single particle and the large ensemble of nanoparticles as well. These facts attest to the notion that upconversion properties of single particles are highly individual. This implies that to use an upconversion particle as a single sensor of the local parameters of a medium, the additional studying and calibration of its individual photophysical properties are essential.

## 1. Introduction

It is very well established that a pair of rare-earth ions Yb^3+^-Re^3+^ (Re = Er, Tm, Ho) exhibits excellent upconversion capabilities [1,2,3]. The mechanism is as follows. After Yb^3+^ ions absorb near-infrared photons, the cross-relaxation process accounts for the energy transfer to the nearest Re^3+^ ions. Two consecutive acts of Yb^3+^ → Re^3+^ energy transfer populate the upper levels of Re^3+^ ions, and the following radiative relaxation leads to a bright green upconversion luminescence. The Yb^3+^-Re^3+^ systems are much more efficient in comparison to the two-photon absorption and second harmonic generation because of real intermediate and high-lying excited levels of Re^3+^ ions [2,4].

Due to the rapid increase in the development of nanoscale technologies during the last decade, upconversion nanoparticles (UCNPs) attract much attention as potential nanofluorophores for a broad range of applications. They have already been successfully tested in experiments with information storage [5], volumetric displays [6], photoactivation chemistry [7] and photovoltaics [8]. Since excitation spectrum peaked at 980 nm wavelength falls into so-called transparency window of biological tissue, utilization of UCNPs in biomedical field [9], this has opened up wide perspectives in bioimaging [10], biodetection [11], theranostics [12], optogenetics [13], drug delivery [14], etc.

Mostly, the mentioned above applications rely on upconversion properties of large nanoparticle ensembles (LNPEs). However, using a single UCNP expands opportunities for achieving nanoscale resolution. Potentially, this approach allows one to measure the effect on local parameters of the environment. For example, the challenging task of low-invasive monitoring of intracellular temperature [15,16] requires employing very few (ideally single) fluorescent nanosensors.

Unfortunately, measuring the luminescence response of a single UCNP requires far more extensive and challenging experimental efforts as compared to an LNPE [17]. This could, possibly, explain why there is not much information about photophysical properties of a single UCNP in the literature. Nevertheless, it is quite understandable that many factors (such as individual composition, lattice defects, ionic distribution, size, surface status and so on and so forth) have a great impact on upconversion properties [18,19].

In this work, we are focused on studying upconversion characteristics of a single particle (SP) in comparison with an LNPE. The choice of YVO_4_:Yb, Er nanoparticles as an object was determined by several considerations. Firstly, the synthesis method is well described in literature and easily implemented [20,21]. Secondly, the upconversion luminescence of the YVO_4_:Yb, Er LNPE is well documented [22,23,24]. Finally, since the yttrium vanadate nanoparticles are not very sensitive to surface quenchers in the water solution, they are of special interest in biological applications. For example, YVO_4_:Yb, Er UCNPs are highly promising as low-invasive upconversion sensors for biological tissues [21,25,26].

According to the reports [17,27,28,29], single UCNPs exhibit polarized emission. Even spectral components of the emission corresponding to the transitions between different energy levels (for example, red and green bands in the luminescence spectrum [30]) could have quite distinguishable polarization characteristics. Evidently, this phenomenon reflects the properties of crystalline order and considerable local crystal field impact on the intensities and polarizations of the emission transitions that is of great interest to spectroscopic studies. On the other hand, the luminescence polarization is a considerable negative factor for employing SPs as single nanosensors, making the luminescence response dependent on SPs’ orientations.

With the aim to avoid or decrease a luminescence polarization degree, we intended to test a relatively large (of 1-μm size) and essentially formless SP. If the upconversion properties of the selected SP would appear to be close to those of the LNPE, then taking into account that a 1-μm size provides relatively intense upconversion luminescence as compared to 10–100 nm SPs, the selected SP might be considered as the ready-to-use biosensor.

Having examined the upconversion properties of the synthesized LNPE (to be assured of their suitability), we applied methods of atomic-force microscopy (AFM) to clean up the contaminants off the glass surface, and then selected the required SP and placed it in the center of the cleaned area. These preparations allowed us to study the upconversion characteristics of the selected SP using conventional confocal optical microscopy (COM), eliminating possible interference from other luminescent particles.

## 2. Experimental Section

### 2.1. Hydrothermal Synthesis of YVO_4_:Er, Yb UCNPs

UCNPs were prepared according to the following procedure [20,21,22,23,24]. A solution of Y(NO_3_)_3_, Er(NO_3_)_3_ and Yb(NO_3_)_3_ (c = 0.1, 0.002 and 0.02 mol/L, respectively) was slowly added to an Na_3_VO_4_ solution (c = 0.1 mol/L) under constant stirring at room temperature. A white precipitate, corresponding to crude YVO_4_:Yb, Er NPs, was obtained and further purified by dialysis. A silica sol was prepared by heating tetraethylorthosilicate, ethanol and distilled water at pH = 1.25, T = 60 °C for 1 h. Crude yttrium orthovanadate particles were then incorporated into silica sol with a dispersing polymer (PE6800) (the molar ratio of V/Si/PE6800 = 1:5:0.05). After drying, a mesoporous silica network was obtained, encapsulating the NPs. It was calcinated at 500 °C for 1 h and then annealed at 1000 °C for 10 min. The silica matrix was removed by a 3 h treatment in hydrofluoric acid with the molar ratio of HF/Si = 9:1.

### 2.2. Characterization of the Large Nanoparticle YVO_4_:Er, Yb Ensemble (LNPE)

Figure 1 presents the surface morphology of the synthetized UCNPs. The image was made by using a scanning electron microscope (SEM) “EVO 50 XVP” (Carl Zeiss, Jena, Germany) equipped with “INCA Energy-350” (Oxford Instruments, Abingdon, UK).

Fluorescence excitation measurements were carried out using a Ti:Sapphire CW laser as a tunable pump source. The intensity of the laser was tuned by a neutral density filter to deliver 5 mW at every excitation wavelength into the sample chamber of a FluorologQM spectrofluorometer. The luminescence of LNPE in the form of a dry powder was detected using an emission channel of the FluorologQM spectrofluorometer, equipped with a 360 mm double monochromator (1200 g/mm grating) and an R13456-11 thermoelectrically cooled photomultiplier tube. Intensity dependent measurements were carried out using a 980 nm CW diode laser equipped with absorption and interference (3 nm FWHM) filters. The pump laser beam was focused on the sample on a 3 mm spot.

### 2.3. Atomic-Force Microscope (AFM) Preparation of Sample for SP Spectroscopy

The study of the surface requires precise positioning in order to obtain access to each particle. Therefore, grid lines (Figure 2a) in the form of straight micro-scratches were made on the glass substrate prior to the UCNP’s deposition. It is worth noting that the grid was essential in studying the selected SP by either AFM (Figure 2c), SEM (Figure 2a,b) or COM (Figure 2d) methods.

The solver-Bio (NT–MDT) atomic-force microscope was used for the characterization and manipulation of nano-objects. The AFM operates in scanning-by-probe mode; the maximum scanning area is about 95 × 95 µm^2^. The setup includes an optical microscope (Biolam) operating in the transmission mode with positioning of the AFM probe on transparent substrates with an accuracy of several µm. N11-A AlBS AFM-probes of 3 N/m stiffness and 60 kHz resonant frequency were used. The probes support both contact and semi-contact modes. We have applied the contact mode to move nano-objects (which are UCNPs and contaminants) outside the study area. The test of cleaning quality and the control of the SP positioning were performed in the semi-contact mode.

Taking into account the weak SP’s adhesion to the glass surface, the following algorithm was applied to prepare the sample with the SP. Initially, an area of about 60 × 60 µm^2^ located near the micro-scratch’s intersection was cleaned up from all objects by using the AFM in contact mode. After cleaning, the AFM-probe was brought in contact with the surface and oscillated in a broad frequency range with a high magnitude. Then, we had selected one of the upconversion microparticles and moved it to the center of the cleaned area for further studying. Finally, a control scan in semi-contact mode was carried out to make sure that only one particle was left in the area of interest (see Figure 2).

The AFM image (Figure 2c) confirms that the selected microparticle is single; its height is about 630 nm (see Appendix A, Appendix A). The AFM image is in agreement with data on the lateral shape and size of microparticles obtained by SEM (Figure 2b).

### 2.4. Luminescence Response of the Single Particle (SP)

Luminescent properties of the SP were studied using a homemade optical confocal microscope. The UCNP luminescence was excited by a MicronLux laser at 980 nm and the maximum power of 100 mW. The pump beam was focused by means of a 40× objective (NA = 0.70) into a spot with a 1-μm waist diameter. For lateral scanning, a pair of galvo mirrors was used. The luminescence radiation was collected by the objective from the upper surface of the sample, where the SP was located. A FELH-700 filter was used to separate the luminescence from the pump radiation. The collected luminescence passed through the pinhole and was divided into two parts by a 50/50 beam splitter. The first part was registered by the photon counter, while the second part was used to record a luminescence spectrum by means of a CCD camera (Starlight Xpress Trius SX-694, Starlight Xpress Ltd., Berks, UK) equipped with a diffraction grating (300 g/mm).

First, the substrate was scanned without the pump cut-off filter at low pump laser power, obtaining a reflected laser radiation map in order to find the cleaned area and locate the SP (Figure 2c, background image). The second scanning with the installed FELH-700 filter provided SP’s luminescence response. (Figure 2d, inserted frame “F”). The luminescence dependence on the pump power was recorded by varying the current through a laser diode and controlling the power of the laser irradiation incident on the sample. We studied the polarization characteristics of the SP emission by using a polarization analyzer in front of the pinhole. As a result, the dependence of the emission spectra on the analyzer angle, varied with a step of 10 degrees, was registered.

## 3. Results

### 3.1. Large Nanoparticle YVO_4_:Yb, Er Ensemble (LNPE)

The scanning electron microscope image of the YVO_4_:Yb, Er LNPE sample, constituting clusters of dry powder on a glass substrate, is presented in Figure 1. Figure 3 shows the upconversion green luminescence of the LNPE under the laser irradiation at 980 nm. The emission bands in the range of 520–570 nm are due to the radiative intra-ionic 4f-4f transitions from the ^2^H_11/2_ and ^4^S_3/2_ levels to the ^4^I_15/2_ ground state of Er^3+^ ions. The excitation spectrum peaked at 980 nm proves that the Er^3+^ luminescence is sensitized by Yb^3+^ ions, providing an efficient Yb^3+^ → Eb^3+^ energy transfer.

The change in the relative intensity of the band peaked at 554 nm is shown (Figure 4) to be influenced by the laser power variation. The integral intensity of the Er^3+^ luminescence exhibits nonlinear dependence on the laser power (see Figure 5): the slopes are 1.6 and 1.4 for the 525 and 554 nm bands, respectively. Thus, the multiphoton character of the Er^3+^ excitation in the synthesized nanoparticles is indisputable. Such upconversion behavior of YVO_4_:Yb, Er LNPEs is well documented in literature and confirms that our synthesized UCNPs are effective upconversion systems [20,21,22,23,24].

### 3.2. Single YVO_4_:Yb, Er Particle (SP) on the Cleaned Glass Substrate

The noticeable dependence of the YVO_4_:Yb, Er SP upconversion emission spectrum on the laser power can be observed in Figure 6. Besides the strong dependence of the overall integral intensity of the Er^3+^ luminescence, one can notice the considerable changes in the relative integral intensity of the band peaked at 554 nm. Interestingly, the dependencies for the SP presented in Figure 7 are quite different from those of the LNPE in Figure 5. In the case of the SP, the signals reach their maximum at some level of the laser power and then decay with the further increase in laser power.

Polarized emission is yet another distinctive feature of the YVO_4_:Yb, Er SP as compared with the LNPE. Figure 8 shows the dependence of the Er^3+^ emission spectrum on the analyzer angle. To avoid experimental errors due to the change in the precise optical geometry of the confocal scheme with the variation of the analyzer angle, normalized integral intensities are presented. The dependence of the relative intensities of the bands peaked at 525 and 554 nm on the analyzer angle is evident. At least, this observation indicates a difference in the polarizations of the emission transitions from the ^2^H_11/2_ and ^4^S_3/2_ levels of Er^3+^ ions. Meanwhile, for the LNPE, no polarization effects were detected.

## 4. Discussion

The slopes presented in Figure 5 clearly point out the nonlinear character of the detected upconversion luminescence. As is known, the slope value reflects the efficiency of the non-radiative relaxation of the Er^3+^ ion levels: its value increases with the rate of multiphonon transitions, quenching the emitting ^2^H_11/2_ and ^4^S_3/2_ levels [31,32]. For example, dispersing the YVO_4_:Yb, Er nanoparticles in water solutions leads to slight luminescence quenching with the increase in the slope values [21,24,25,26]. In our case, the relatively small values imply a low efficiency of multiphonon transitions if compared to the reported data [20,21,22,23,24].

Another fact indirectly confirming the low efficiency of the non-radiative relaxation is the absence of a noticeable red emission band at 660 nm. The intensity of this band is proportional to the population of the ^4^F_9/2_ level of Er^3+^. There are two channels of populating the ^4^F_9/2_ level: the first one is the downward transition from the upper Er^3+^ level ^4^S_3/2_ → ^4^F_9/2_; the second one consists of the downward transition ^4^I_11/2_ → ^4^I_13/2_ and the following inter-ionic Yb^3+^ → Eb^3+^ energy transfer: Yb^3+^ ions, ^2^F_5/2_ → ^2^F_7/2_; Er^3+^ ions, ^4^I_13/2_ → ^4^F_9/2_. Since both channels include multiphonon transitions, the low population of the ^4^F_9/2_ level suggests that the multiphonon mechanism is not efficient.

It is interesting to discuss the observed differences in the laser power dependencies of the band intensities in Figure 5 and Figure 7 for the LNPE and the SP. Although the selected SP is one of the largest particles with the brightest upconversion luminescence among the synthesized particles, it is important to note that detecting the luminescence signal with the reasonable signal-to-noise ratio was possible for the laser intensity exceeding a threshold of about 200 W/mm^2^. In order to measure the laser power dependence shown in Figure 7, the laser intensity was varied in the range of 240–2400 W/mm^2^. For comparison, the laser intensity in the range of 5–35 mW/mm^2^ was sufficient to observe the changes in the luminescence spectrum of the LNPE presented in Figure 5.

The presented experimental results undoubtedly reveal an inequality between the photophysical properties of the LNPE and SP samples. It has to be noted that the reliable detection of the upconversion luminescence of the SP requires the laser intensity of several orders higher as compared to the LNPE. The intrinsic inequality in the experimental conditions therefore does contribute to the observed difference in the upconversion properties of the LNPE and the SP.

At least three factors responsible for the observed laser power dependence of the SP luminescence could be noted. The saturation effect at a certain level of the laser irradiation occurs [33,34,35] when the rate of inter-ionic Yb^3+^ → Eb^3+^ energy transfer is limited due to the depletion of excited intermediate states, which also limits the population of the ^2^H_11/2_ level and, accordingly, the 525 nm emission intensity. The next factor is the well-known temperature sensitivity of the ^2^H_11/2_ and ^4^S_3/2_ levels populations of Er^3+^ [23,36,37,38,39]. After reaching the saturation intensity, a further increase in the laser power continues to heat up the SP. In such conditions, the thermal activation process speeds up the upward transition ^4^S_3/2_ → ^2^H_11/2_ that could decrease the ^4^S_3/2_ level population, revealing itself as the observed decrease in the integral intensity of the 545 nm band in the range of 4.5–8 mW pump power (see Figure 7). As the third possible factor, a known multiphonon mechanism contributing to the population distribution among the levels of Er^3+^ ions [40] can be considered. Since the temperature rise increases the rates of cascade downward multiphonon transitions, this should result in the decrease in the population of the upper levels.

Since special efforts were made to select one of the biggest and formless upconversion particles for the SP sample in the anticipation that its luminescent characteristics would be close enough to those of the LNPE’s, the observed polarization dependence presented in Figure 8 was quite surprising. First, such an effect implies different polarizations of the ^2^H_11/2_ → ^4^I_15/2_ and ^4^S_3/2_ → ^4^I_15/2_ emission transitions. Although the polarized upconversion emission of single nanoparticles has not been fully studied, according to the reports [17,27,41], the polarizations of emission transitions could be very sensitive to the crystal structure features. Second, we have to assume that the averaged crystal structure anisotropy of the SP is non-zero. The formless SP in Figure 2 apparently has some crystalline order, which is sufficient to make the polarizations of the ^2^H_11/2_ → ^4^I_15/2_ and ^4^S_3/2_ → ^4^I_15/2_ emission transitions quite distinguishable.

The current growing tendency to use UCNPs as nanosensors in a broad range of applications implies monitoring the local medium parameters with a small amount of UCNPs or even by using a single UCNP (for example, for low-invasive measurements of intracellular temperatures). Our results suggest that calibration made with the LNPEs will hardly reflect the luminescence parameters of SPs because of their individuality. It means that using a SP or a small ensemble of UCNPs as nanosensors requires their individual calibration.

## 5. Conclusions

YVO_4_:Yb, Er nanoparticles were synthesized by using the hydrothermal method. Our study focused on the difference between the upconversion luminescence of a single particle (SP) and a large ensemble of nanoparticles (LNPE). We started with the examination of the LNPE, which exhibited the ordinary photophysical properties well described in the literature. Then, we selected the brightest, largest and seemingly formless SP and placed it in the center of the cleaned area of a glass substrate in the anticipation that its upconversion characteristic would not deviate much from those of the LNPE. However, the performed measurements clearly indicated that the detection of the luminescence signal requires the laser power density of several orders higher as compared with the LNPE. Moreover, the variation of the laser power revealed the saturations effect. The nonequivalent experimental conditions hinder the comparative analysis of the SP and LNPE luminescence characteristics. The interesting and quite unexpected observation of the polarization dependence of the Er^3+^ emission was made. Despite the formless character and the relatively big size of the SP, we detected a pronounced change in luminescence intensity with the variation of the analyzer angle. Thus, we have to conclude that (i) the polarizations of the ^2^H_11/2_ → ^4^I_15/2_ and ^4^S_3/2_ → ^4^I_15/2_ emission transitions are different; (ii) the SP has some degree of crystalline order, making the polarizations of these two transitions distinguishable. The revealed facts reflect the fundamental non-equivalence between SPs and LNPEs primarily due to SP’s individuality. Since SPs are considered as a decent platform for monitoring and controlling the parameters of local environments in a broad range of applications, the presented results provide relevant points to consider when developing precise nanosensing technologies.

## Figures and Tables

**Figure 1 micromachines-14-01075-f001:**
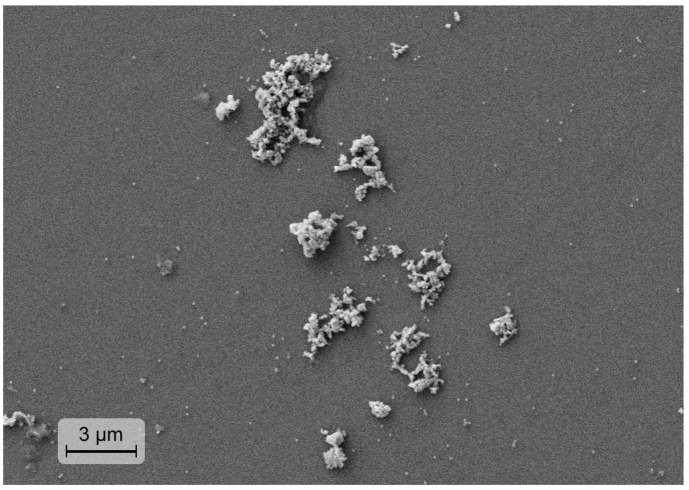
Scanning electron microscope image of the dry YVO_4_:Yb, Er nanoparticle clusters on a glass substrate.

**Figure 2 micromachines-14-01075-f002:**
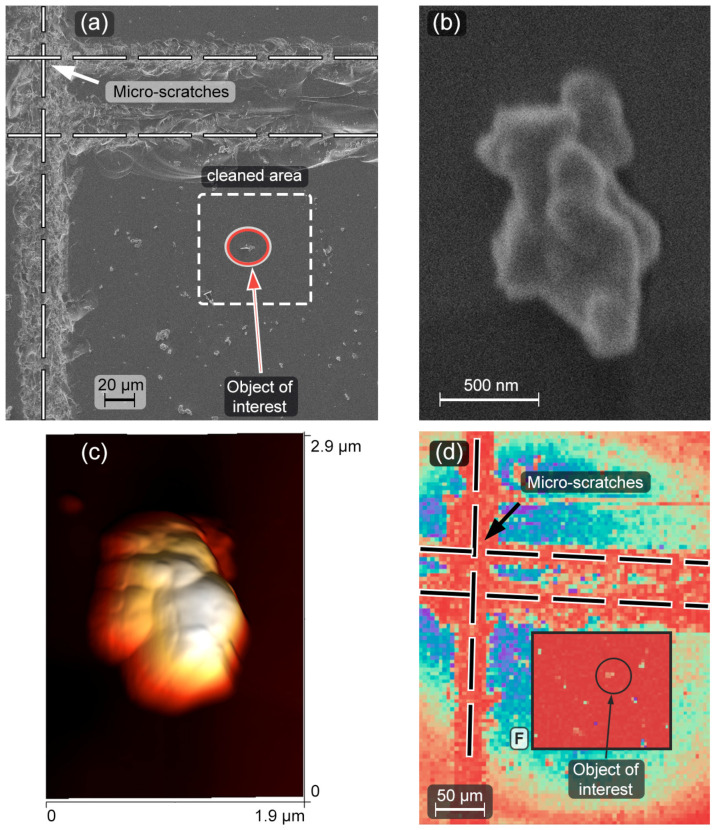
SEM image of grid lines (micro-scratches) made on the glass substrate and location of the cleaned area (**a**), SEM (**b**) and 3D-AFM (**c**) images of SP. COM image (**d**) of the sample in reflection mode with highlighted insert “F” of the cleaned area in luminescence mode, the cleaned area and the SP are clearly visible.

**Figure 3 micromachines-14-01075-f003:**
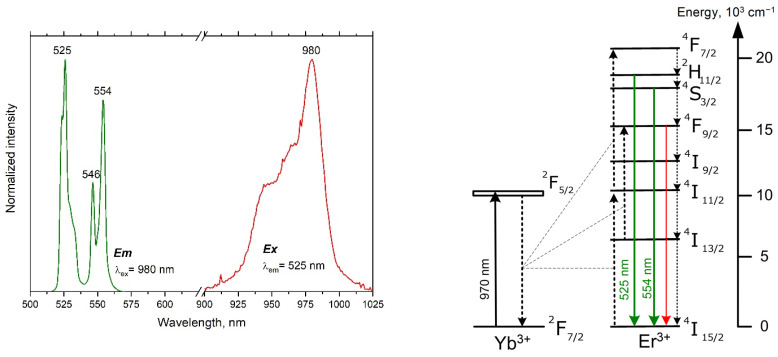
Steady-state luminescence (green) and excitation (red) spectra of the YVO_4_:Yb, Er LNPE (**left**). The energy level diagram for the Yb^3+^-Er^3+^ upconversion system (**right**). Solid arrows are radiative transitions; dotted arrows depict cross-relaxation energy transfers and multiphonon relaxation channels.

**Figure 4 micromachines-14-01075-f004:**
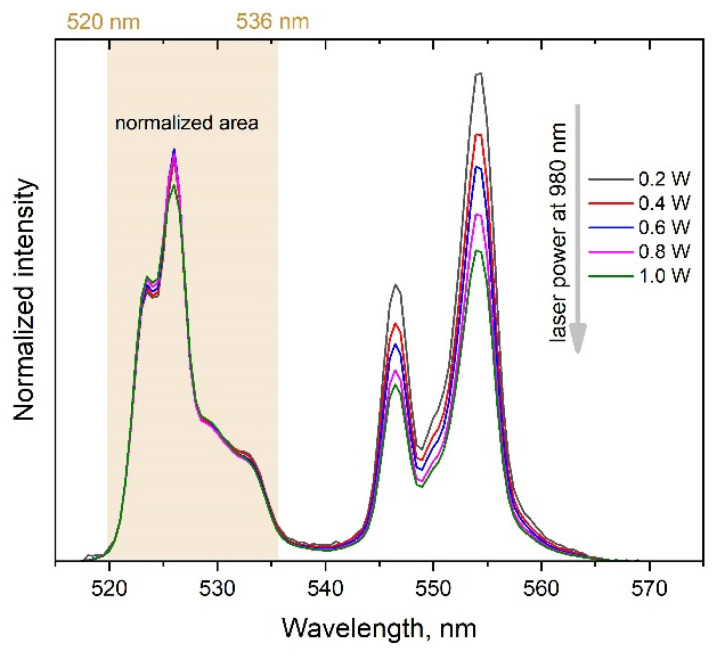
Dependence of the Er^3+^ emission spectrum on the pump laser power of the YVO_4_:Yb, Er LNPE. Spectra are normalized to the integral intensity of the 520–536 nm band.

**Figure 5 micromachines-14-01075-f005:**
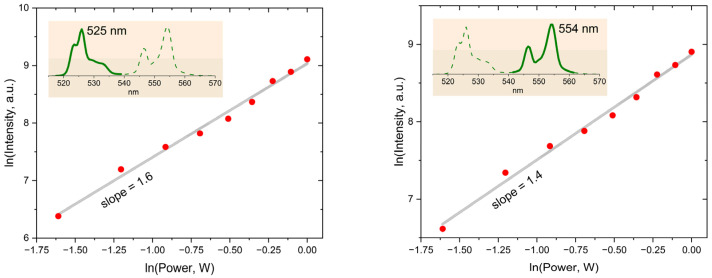
Dependences of the integral intensities of the Er^3+^ luminescence in the 520–540 nm (**left panel**) and 540–563 nm (**right panel**) ranges on the laser power measured for the YVO_4_:Yb, Er LNPE. Red dots are experimental data, grey straight line is power fitting function, green lines are Er^3+^ luminescence spectrum, where green solid line indicates integrating spectral region.

**Figure 6 micromachines-14-01075-f006:**
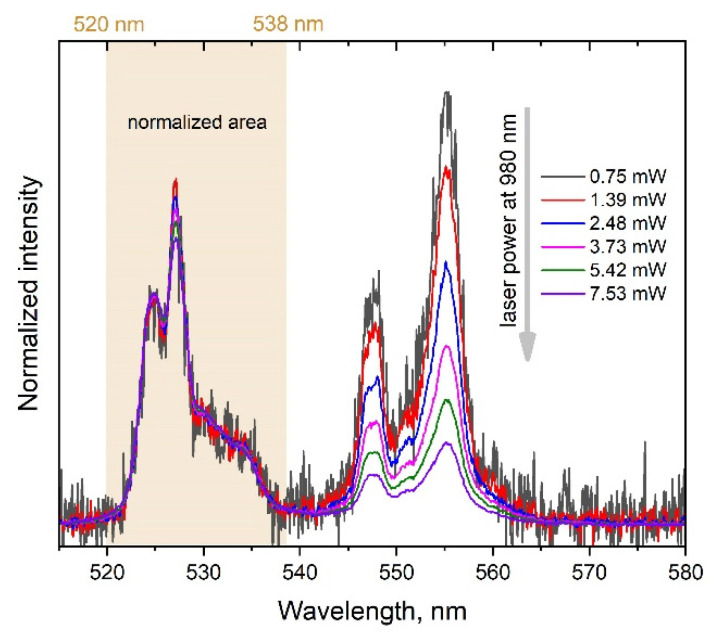
Dependence of the Er^3+^ emission spectrum on the laser pump power measured of the YVO_4_:Yb, Er SP. Spectra are normalized to the integral intensity of the 520–536 nm band.

**Figure 7 micromachines-14-01075-f007:**
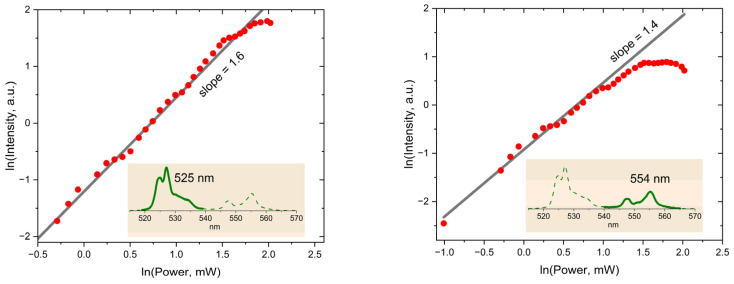
Dependencies of the integral intensities of the Er^3+^ luminescence in the 520–540 nm (**left panel**) and 540–565 nm (**right panel**) ranges on the laser power of the YVO_4_:Yb, Er SP. Red dots are experimental data, grey straight line is power fitting function, green lines are Er^3+^ luminescence spectrum, where green solid line indicates integrating spectral region.

**Figure 8 micromachines-14-01075-f008:**
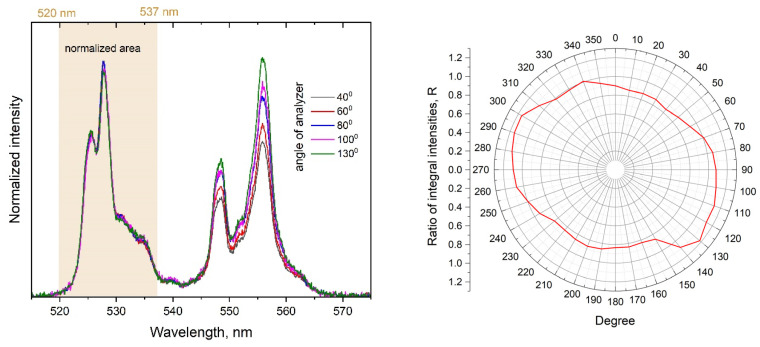
The dependence of the Er^3+^ spectrum normalized to the integral intensity in the range of 520–537 nm on the analyzer angle (**left**). Red line is the change in the ratio $R=\int_{520\;\mathrm{nm}}^{537\;\mathrm{nm}}I\left(\lambda \right)d\lambda /\int_{545\;\mathrm{nm}}^{562\;\mathrm{nm}}I\left(\lambda \right)d\lambda$ with the analyzer angle, where $I\left(\lambda \right)$ is spectral intensity in arbitrary units (**right**).

## Data Availability

The data presented in this study are available on request from the corresponding author.

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
