# Peer review of "Upconversion Luminescence Response of a Single YVO4:Yb, Er Particle"

_micromachines, 2023, doi:10.3390/mi14051075_

Round 1

Reviewer 1 Report

The authors analyze the optical properties of YVO4: Yb, Er comparing a single particle of 1 micron size with a large ensemble. The analysis is based on the upconversion emission obtained in both experiments. But the main problem is the power range used for excitation at 980 nm.  I have several issues or comments:

-          For the ensemble the authors used laser intensity in the range of 5-35 mW/mm2 while for the single nanoparticle the range was 240-2400 W/mm2. Maybe there is a mistake in the units.

-          The authors compare the dependence of the upconversion emission but the figures (Figs. 5 and 7) are plotted in different format (logarithmic and linear). Moreover, is not possible to compare both experiments if the heating effect is very important because, as can be seen in the Figures 5 and 7, the ratio of the emissions at 525 and 550 change appreciably.

-          I agree with the authors and their results. It is expected an important change with the polarization in small samples, but the changes detected and shown in Fig. 8 don’t look reproducible. Do the authors have repeated the experiment with other single particle?

-           

Author Response

The authors analyze the optical properties of YVO4: Yb, Er comparing a single particle of 1 micron size with a large ensemble. The analysis is based on the upconversion emission obtained in both experiments. But the main problem is the power range used for excitation at 980 nm.  I have several issues or comments:

-          For the ensemble the authors used laser intensity in the range of 5-35 mW/mm2 while for the single nanoparticle the range was 240-2400 W/mm2. Maybe there is a mistake in the units.

Answer

Unfortunately, there is no mistake. Measuring the luminescence signal of a single particle requires much more intense laser excitation in comparison to a large ensemble. That is not our experimental specifics, other researches deal with the same problem, see, for instance, a review [Nano Today 35, 100956 (2020)].

-          The authors compare the dependence of the upconversion emission but the figures (Figs. 5 and 7) are plotted in different format (logarithmic and linear). Moreover, is not possible to compare both experiments if the heating effect is very important because, as can be seen in the Figures 5 and 7, the ratio of the emissions at 525 and 550 change appreciably.

Answer

In the revised manuscript, we provide the logarithmic format for both Figs. 5 and 7 to make the comparison more clear.

-          I agree with the authors and their results. It is expected an important change with the polarization in small samples, but the changes detected and shown in Fig. 8 don’t look reproducible. Do the authors have repeated the experiment with other single particle?

Answer

Thank you for the interesting comment. Indeed, it would be important to compare properties of different single particles, and we intend to do it. However, it is going to take a lot of time and experimental efforts to isolate and investigate a set of individual particles. Since, at the moment, we have finished experiments with the one single particle, we decided to report our observations. Based on results in the manuscript, we believe that the changes shown in Fig. 8 are not reproducible with  other single particles due to their uniqueness. Of course, has to be verified.

Reviewer 2 Report

This work is focused on the spectral analysis of upconversion in a single YVO4:Yb,Er particle in comparison to an ensemble of several nanoparticles. The subject is interesting, but some aspects are not convincing.

(1)          The authors compare the emission of a single 1 micron particle with that of an ensemble of nanoparticles. It would have been more correct to use same-size objects for comparing the difference between a single element and an ensemble of elements.

(2)          Figure 5 should report the ln(Intensity) vs. ln(Power) for the correct evaluation of the slope factor. Please check this issue. Figure 7 should also report the same logarithmic presentation, not a linear-linear graph, to compare the fitting coefficients with the ones of Figure 5. In particular, the slope in Figure 7 (right panel) seems linear.

(3)          Similar power ranges should be explored to discuss about the saturation effect. The power used for the single particle is much higher than that used for the nanoparticle ensemble and this can be one of the reasons for the observed saturation.

(4)          Finally, this paper is focused on the results obtained on one specific single particle. This is very critical. I would strongly recommend to have at least a confirmation of what observed on other single particles.

English 

Author Response

This work is focused on the spectral analysis of upconversion in a single YVO4:Yb,Er particle in comparison to an ensemble of several nanoparticles. The subject is interesting, but some aspects are not convincing.

(1)          The authors compare the emission of a single 1 micron particle with that of an ensemble of nanoparticles. It would have been more correct to use same-size objects for comparing the difference between a single element and an ensemble of elements.

Answer

It is a very appealing idea to make comparison between a single unit and an ensemble of the same units. It should be noted that obtaining an ensemble of identical (or very close to identical) nanoparticles is in itself a quite difficult problem, because simple synthesis methods as a rule provide the nanostructures with large size dispersion (and, consequently, having large difference in their photophysical properties). In the presented work, we test the completely opposite case: what is the difference between one of most large and formless particles and an ensemble of differently sized particles (without any separation made post-synthesis)? For example, if the properties of such particle would be close to the properties of an ensemble, that would mean that the particle somehow combines all the cases of ion distributions in lattices, defects and so on inherent to the whole ensemble.

(2)          Figure 5 should report the ln(Intensity) vs. ln(Power) for the correct evaluation of the slope factor. Please check this issue. Figure 7 should also report the same logarithmic presentation, not a linear-linear graph, to compare the fitting coefficients with the ones of Figure 5. In particular, the slope in Figure 7 (right panel) seems linear.

Answer

We have fixed it. In the revised manuscript, both Figs. 5 and 7 are presented in the logarithmic format.

(3)          Similar power ranges should be explored to discuss about the saturation effect. The power used for the single particle is much higher than that used for the nanoparticle ensemble and this can be one of the reasons for the observed saturation.

Answer

We absolutely agree that it would be much better to use the same power range in both cases. However, it is technically impossible with our current experimental set up. Reliable single particle luminescence detection is possible with excitation intensity no less than 200 W/mm^2. On the other hand, the conditions of the experiments on large nanoparticle ensemble require 1 – 3 mm laser spot and a quite powerful cw laser to provide similar intensities. It should be noted that these kinds of experimental limitations are very common in single particle spectroscopy [for example, Nano Today 35, 100956 (2020)].

(4)          Finally, this paper is focused on the results obtained on one specific single particle. This is very critical. I would strongly recommend to have at least a confirmation of what observed on other single particles.

Answer

We absolutely agree that comparison with other single particles is of great importance. We intend to continue this study by repeating the experiments with other particles. That will however take some additional time and experimental efforts. At this moment, we have analyzed the experimental data for only one single particle, which still find to be valuable.

Reviewer 3 Report

The manuscript presents the upconversion luminescence response of single YVO4:Yb,Er particle (SP) excited with a 980 nm laser. The polarization effect, excitation power density, and saturation effect of SP were compared with those of large nanoparticle YVO4:Yb,Er (LNPE). The elucidation of SP upconversion luminescence is clear and reasonable, although the difference in luminescence response between SP and LNPE still has room for investigation.

Some information given in the text may need to be revised:

1.     Line 164, in the description “…luminescence response (Fig. 2c, inserted frame "F")…”, should “Fig. 2c” be “Fig. 2d”?

2.     In the energy level diagram of Fig. 3, should “550 nm” be “554 nm”?  

Author Response

The manuscript presents the upconversion luminescence response of single YVO4:Yb,Er particle (SP) excited with a 980 nm laser. The polarization effect, excitation power density, and saturation effect of SP were compared with those of large nanoparticle YVO4:Yb,Er (LNPE). The elucidation of SP upconversion luminescence is clear and reasonable, although the difference in luminescence response between SP and LNPE still has room for investigation.

Some information given in the text may need to be revised:

  1. Line 164, in the description “…luminescence response (Fig. 2c, inserted frame "F")…”, should “Fig. 2c” be “Fig. 2d”?

Answer

It is fixed in the revised manuscript.

  1. In the energy level diagram of Fig. 3, should “550 nm” be “554 nm”? 

Answer

It is fixed in the revised manuscript. 

Reviewer 4 Report

Article review Upconversion luminescence response of a single YVO4:Yb, Er particle by Dmitry K. Zharkov, Andrey V. Leontyev, Artemi G. Shmelev, Larisa A. Nurtdinova, Anton P. Chuklanov, Niaz I. Nurgazizov and Victor G. Nikiforov

The paper presents the results of studying the luminescence response of single YVO4:Yb, Er particles and an ensemble of particles. Extremely interesting results have been obtained in this work. The difference between the collective luminescent response of an ensemble of synthesized nanoparticles and in the form of a single particle has been established. The luminescence dependences are not comparable for a single particle and a large ensemble of nanoparticles as well. These facts indicate that the up-conversion properties of single particles are very high. Despite the considerable interest in the results of the work, there were some comments.

The article does not characterize the obtained nanoparticles. It is not clear how the authors of the work prove that they are dealing with doped yttrium vanadate. It is also not clear what the concentration of dopants in these systems is. Is it possible to somehow estimate the quantitative composition of the resulting systems? The photomicrograph given by the authors is fuzzy. Is it possible to get a high magnification photomicrograph?

Author Response

Article review Upconversion luminescence response of a single YVO4:Yb, Er particle by Dmitry K. Zharkov, Andrey V. Leontyev, Artemi G. Shmelev, Larisa A. Nurtdinova, Anton P. Chuklanov, Niaz I. Nurgazizov and Victor G. Nikiforov

The paper presents the results of studying the luminescence response of single YVO4:Yb, Er particles and an ensemble of particles. Extremely interesting results have been obtained in this work. The difference between the collective luminescent response of an ensemble of synthesized nanoparticles and in the form of a single particle has been established. The luminescence dependences are not comparable for a single particle and a large ensemble of nanoparticles as well. These facts indicate that the up-conversion properties of single particles are very high. Despite the considerable interest in the results of the work, there were some comments.

The article does not characterize the obtained nanoparticles. It is not clear how the authors of the work prove that they are dealing with doped yttrium vanadate. It is also not clear what the concentration of dopants in these systems is. Is it possible to somehow estimate the quantitative composition of the resulting systems?

Answer

Thank you for this critical and valuable comment. We have chosen a well-described, proven and quite simple synthesis method [ACS Nano 2, 2505–2512 (2008), J. Phys. Chem. C 114, 22449–22454 (2010), Laser Phys. Lett. 17, 075901 (2020), J. Opt. Soc. Am. B 31, 1814 (2014)]. According to the method, the concentrations c = 0.1, 0.002, and 0.02 mol/l of the solution components Y(NO3)3, Er(NO3)3, and Yb(NO3)3, respectively, result in yielding vanadate oxide nanoparticles YVO4:20%Yb,2%Er. We are conscious that dopants concentrations of the selected single particle could be quite different compared to average concentrations over the large nanoparticle ensemble. With the aim to estimate the quantitative composition, we have performed the SEM EDS analysis of the single particle. According to the data presented in Supplementary Material, the concentration ratio is about Yb/Er ≈ 3/1. This is very raw estimation, indicating (i) the presence of both Yb^3+ and Er^3+ ions and (ii) Yb^3+ concentration exceeding the one of Er^3+. Based on this result, it is possible to assume that dopants’ concentrations in the single particle are not far away from those of the large nanoparticle ensemble YVO4:20%Yb,2%Er.

The photomicrograph given by the authors is fuzzy. Is it possible to get a high magnification photomicrograph?

Answer

We improved the quality of the AFM image in Fig. 2c. Additional AFM data are presented in Supplementary Material.

Round 2

Reviewer 2 Report

The authors clarified the critical points. Some of them could not be changed due to experimental limitations, but the overall message is now more clear.